# Double-Inlet Left Ventricle

**DOI:** 10.3390/children9091274

**Published:** 2022-08-24

**Authors:** P. Syamasundar Rao

**Affiliations:** Children’s Heart Institute, Children’s Memorial Hermann Hospital, McGovern Medical School, University of Texas-Houston, Houston, TX 77030, USA; p.syamasundar.rao@uth.tmc.edu or srao.patnana@yahoo.com; Tel.: +1-713-500-5738; Fax: +1-713-500-5751

**Keywords:** double-inlet left ventricle, single ventricle, Blalock–Taussig anastomosis, banding of the pulmonary artery, inter-stage mortality, bidirectional Glenn operation, Fontan surgery

## Abstract

Double-inlet left ventricle (DILV) is most frequent among univentricular atrioventricular connections. In DILV, there is a single functioning ventricle, most commonly with left ventricular structure. This chamber receives both atrioventricular valves and is connected to an outlet chamber with morphologic features of the right ventricle. The great vessels are often transposed, and pulmonary stenosis is seen in two-thirds of patients. The anatomy and pathophysiology can be defined by echo-Doppler studies with a rare need for other imaging studies. The management is mostly related to the nature of associated heart defects and the degree of pathophysiological abnormality. When the infants present initially, treatment to address the hemodynamic issues is undertaken. Subsequently, these babies need staged total cavo-pulmonary connection, i.e., the Fontan procedure which is undertaken in three stages; these stages are described in this review. The existence of inter-stage mortality and post-Fontan complications is recognized and was reviewed. The paper concludes that DILV can be successfully diagnosed with echo-Doppler studies and this heart anomaly can be effectively treated with the currently prevailing medical, catheter interventional, and surgical treatment practices.

## 1. Introduction

The term univentricular heart was used to describe any heart defect with one (single) effective ventricular cavity. These heart defects are single ventricle, common ventricle, double-inlet left ventricle (DILV), and univentricular atrio-ventricular connections [1,2,3]. These variations altogether constitute ≤2% of all congenital heart defects (CHDs). In these heart defects, the two atrioventricular valves enter one ventricular chamber, i.e., DILV, one common atrioventricular valve enters the single ventricle or only one of the atrioventricular valves (mitral or tricuspid) enters the single ventricle with atresia of either of the other atrioventricular (mitral or tricuspid) valves. DILV is the most common among the univentricular atrioventricular connections. Hypoplastic left heart syndrome is not included in this review. The aim of this article is to review the current status of the diagnosis and management of DILV. Since the diagnosis and management of the other variants are comparable and the same principles apply, they will not be reviewed separately.

## 2. Pathologic Anatomy

In patients with DILV, the main ventricular cavity usually exhibits left ventricular morphology [1,2,3]. However, other ventricular morphologies, namely, right ventricle, mixed, indeterminate, or undifferentiated variants have been described previously. The main ventricular chamber is primarily a morphologic left ventricle. An outlet chamber with morphologic characteristics of the right ventricle is attached to it. Both the atrioventricular valves are usually normal. However, one of the atrioventricular valves may be atretic, hypoplastic, or stenotic. In another variant, a common and single atrioventricular valve exits into the single ventricle. Transposition of the great arteries is frequently seen with the aorta arising from the hypoplastic right ventricle and the pulmonary artery coming off the main left ventricular chamber. l-transposition of the great arteries (l-TGA) occurs more frequently than d-transposition of the great arteries (d-TGA). The great vessels have normal relationship in nearly 30% of cases. Double-outlet right ventricle with both great vessels coming off the somewhat small-sized right ventricle has also been reported in the literature. Stenosis of the pulmonary valve/outflow tract may be found in two-thirds of patients. Pulmonary stenosis is seen irrespective of the great artery arrangement. The pulmonary outflow obstruction may be at the valvular or sub-valvular level. In some patients, the pulmonary valve and/or artery may be atretic. Obstruction at the subaortic region has been found in subjects with TGA and is associated with the small size of the ventricular septal defect; more correct terminology is bulbo-ventricular foramen (BVF). In the embryo, the BVF connects the bulbus cordis (future right ventricle) with the embryonic ventricle (future left ventricle), thus, justifying use of this term. Cases with subaortic obstruction commonly have coarctation of the aorta. Rarely, aortic arch interruption may also be seen with DILV.

## 3. Pathophysiology

The systemic and pulmonary venous return via the right and left atria, respectively, enter the single ventricle, and this admixture results in reduction in systemic arterial saturation in all patients with DILV [1,2,3]. This mixed blood is then disseminated into the systemic and pulmonary circuits largely based on their respective vascular resistances. In subjects who have stenosis of the pulmonary outflow tract, the severity of pulmonary stenosis determines the magnitude of blood flow into the lungs. In babies who have pulmonary atresia, the blood flow to the lungs is supplied via a patent ductus arteriosus or on occasion via aortopulmonary collateral arteries. In infants with no pulmonary outflow tract obstruction, the pulmonary blood flow is not elevated at, and shortly after birth since the resistance in the pulmonary circuit is high in the neonate. As the baby ages, the vascular resistance in the pulmonary circuit and pressures in the pulmonary artery decrease with successive increase in blood flow to the lungs with ensuing onset of congestive heart failure. Babies with obstructed BVF will experience obstruction of the left ventricular outflow tract in cases with transposition while babies without transposition develop pulmonary oligemia. Aortic coarctation and interruption will impose additional hemodynamic burden to the other pathophysiologic abnormalities.

## 4. Clinical Features

The clinical presentation is mostly determined by the type and severity of associated heart defects, particularly, the severity of pulmonary outflow tract narrowing [1,2,3]. Infants with significant pulmonary stenosis exhibit symptoms of cyanosis and increased respiratory and heart rates (tachypnea and tachycardia). The symptoms of tachypnea and tachycardia are physiologic responses to hypoxemia. Babies with severe pulmonary stenosis manifest very early in the neonatal period, particularly if the ductus arteriosus naturally closes. An increase in ventricular impulse along with a thrill at the upper left sternal border may be felt. Auscultation reveals a single second heart sound and a grade III to IV/VI long systolic ejection murmur along the upper left sternal border. In babies with pulmonary atresia, pulmonary ejection murmurs are not auscultated. It is unusual to detect a patent ductus arteriosus murmur. Findings indicative of heart failure are not seen.

Babies without pulmonary stenosis typically manifest slightly later than those infants with severe pulmonary stenosis or atresia. They manifest with features of congestive heart failure in a few weeks or months after birth. If there is cyanosis, it is usually mild; this is secondary to large blood flow into the lungs. On precordial palpation, the cardiac impulses are increased and hyperdynamic. On auscultation, the cardiac sounds are generally within the normal range. An ejection systolic murmur of grade I to II/VI may be heard at the left sternal border. A mid-diastolic flow murmur at the apical region is auscultated if the pulmonary blood flow is increased. Findings suggestive of heart failure are not observed in the early newborn period but are seen in a few weeks or months later.

Some of these babies may also be detected because of abnormal fetal echocardiographic studies or due to auscultation of a cardiac murmur in the newborn period. Or, they may have been detected because of pulse oximetry screening.

Babies with associated aortic arch obstruction (interrupted aortic arch and aortic coarctation) present early with signs of shock and decreased systemic perfusion as and when the ductus arteriosus constricts.

## 5. Chest X-ray

The chest roentgenographic features vary and largely depend on the quantity of pulmonary blood flow. Slight cardiac enlargement and reduced pulmonary blood flow are observed in children with substantial pulmonary outflow tract obstruction. Cardiomegaly of moderate to severe degree and augmented pulmonary blood flow are observed in infants without pulmonary stenosis. In infants with l-TGA, a prominent and straight left heart border may also be documented.

## 6. Electrocardiogram

There are no electrocardiographic features that are diagnostic of DILV. In some patients, the electrocardiogram may demonstrate an atypical initial QRS vector with a qR pattern in leads V_1_ and V_2_. In other patients, an Rs appearance is perceived in leads V_5_ and V_6_. Yet, some other patients may exhibit right, left, or bi-ventricular hypertrophy, largely dependent on the anatomic type. As mentioned above, none of the electrocardiographic abnormalities are characteristic for DILV.

## 7. Echocardiogram

Echo-Doppler studies, in contrast to the chest X-ray and electrocardiographic findings, are very useful in coming up with a diagnosis and in characterizing pathophysiologic abnormality [2,3]. The lack of the ventricular septum can be demonstrated (Figure 1B, Figure 2B and Figure 3). The connections of the atrioventricular valves (Figure 1B, Figure 2B and Figure 3) can be defined. Similarly, ventriculo–arterial connections (Figure 4A) can be shown. The connection of the smallish right ventricle with the left ventricle via a BVF may be demonstrated by appropriate transducer angulations as shown in Figure 4B. In patients with pulmonary stenosis, its presence is usually documented by two-dimensional and color flow Doppler (Figure 4C) echo studies. Continuous wave Doppler flow velocity recordings through this region will help quantitate the degree of pulmonary stenosis by the use of a modified Bernoulli equation; the higher the Doppler velocity, the greater the degree of obstruction.

In patients with TGA, the dimension of the BVF and signs of obstruction across this region should be examined. Obstruction at the BVF may either be seen at initial presentation, progress subsequently through the normal course of the disease, or may progress subsequent to the placement of the pulmonary artery band [5]. Aortic coarctation (Figure 5) may be seen in some such cases. An interrupted aortic arch may also be demonstrated when imaging the arch of the aorta from the suprasternal notch.

## 8. Other Imaging Studies

Other studies such as computed tomography, magnetic resonance imaging, cardiac catheterization, and selective cineangiography are not needed since the echo-Doppler studies can define all anatomic and pathophysiologic issues related to DILV [2,3]. However, computed tomography and magnetic resonance imaging studies are nowadays routinely performed as a part of the overall evaluation of all complex CHDs at some institutions. Such studies usually confirm the echo findings. An angiographic example of DILV with l-TGA (secured prior to the advent of current echo-Doppler capability) is shown in Figure 6.

## 9. Comparison of Different Diagnostic Methods

Chest X-rays and electrocardiograms are conventional studies performed for evaluation of all CHDs and are valuable in assessing the size of the heart, evaluating pulmonary blood flow and detecting pulmonary pathology, and in detecting arrhythmias. These studies, however, do not supply a diagnosis of the cardiac lesion. On the contrary, echocardiography and Doppler are helpful in coming up with the diagnosis and in assessing pathophysiologic abnormalities and can be performed bedside. Magnetic resonance imaging and computed tomography studies also provide diagnostic information similar to echo studies but require transportation of the baby to a radiology suite, sometimes needing anesthesia. However, these studies can provide information that is not available by echo alone. Cardiac catheterization with selective cineangiography is no longer performed routinely for diagnostic purposes.

## 10. Therapy

Attempts were made initially to insert a prosthetic ventricular septum within the single ventricle [7] and these procedures were unsuccessful and, therefore, such procedures were abandoned and cardiologists/surgeons have searched for other methods of management. Because of the existence of a single functioning ventricular chamber, the general aim is to allow the single ventricular chamber to support the systemic circulation and attach the vena cavae directly to the pulmonary arteries. This concept was originally described by Fontan, Kruetzer, and their colleagues [8,9] in the early 1970s for treating patients with tricuspid atresia. This concept and the procedure were subsequently applied to other cardiac defects with a single pumping chamber including DILV. The types of procedures and when such procedures are performed have progressed over the last few decades, as detailed in our prior publications [10,11,12,13,14,15]. At the present time, the Fontan operation is undertaken by total cavo-pulmonary connection (TCPC), suggested by de Leval [16]. The Fontan operation cannot be accomplished in the newborn and young babies since they have increased pulmonary artery pressures and high pulmonary vascular resistance. Currently, the Fontan procedure is undertaken in three stages [13,14,15] which will be reviewed hereunder.

### 10.1. Stage I

Stage I is the treatment initiated when the baby presents for the first time, usually as a neonate or a young infant and is largely dependent on the pathophysiology of the defect complex and the associated heart defects. Status of blood flow to the pulmonary circuit and the existence of obstructive defects within or outside the heart determine the type of initial therapy.

#### 10.1.1. Reduced Pulmonary Blood Flow

Reduced pulmonary blood flow may be due to atresia or severe pulmonary outflow obstruction. In these babies, the ductus arteriosus must be kept patent by intravenous infusion of prostaglandin E_1_ (PGE_1_); the recommended dose is 0.05–0.1 mcg/kg/min. Once the O_2_ saturations improve, the dosage of PGE_1_ is slowly, step-by-step, reduced to 0.02–0.025 mcg/kg/min in an attempt to decrease the adverse effects of PGE_1_. After the infant is stabilized, further diagnostic studies are performed as necessary. Then, a steadier method of perfusing the pulmonary circuit should be established. Several methods to increase pulmonary blood flow have been employed earlier, as discussed previously [17]. Among such methods, the modified Blalock–Taussig shunt [18], implantation of a stent within the ductus arteriosus [19,20,21], and balloon dilatation of the pulmonary valve (in subjects with major narrowing of the pulmonary valve) [22,23,24] are more commonly used. Nevertheless, most surgeons utilize the modified Blalock–Taussig shunt in which a Gore-Tex graft is inserted connecting the subclavian artery with the pulmonary artery on the same side [18] (Figure 7 and Figure 8) to augment blood flow into the pulmonary circuit.

#### 10.1.2. Elevated Pulmonary Blood Flow

As mentioned in the “Clinical Features” section, a significant increase in blood flow into the lungs results in heart failure. Such babies should first be treated with anti-congestive medications [27]. Regardless of the adequacy of control of the congestive state, banding of the pulmonary artery [28] (Figure 9 and Figure 10) should be performed. This would facilitate successful management of congestive heart failure. In addition, banding normalizes the pulmonary artery pressure and resistance so that the infant can later undergo successful second and third stages of Fontan surgery.

#### 10.1.3. Adequate Pulmonary Blood Flow

Infants with a minimally increased or close to normal pulmonary blood flow with O_2_ saturations in low 80s may not have symptomatology and manifest lower levels of cyanosis than infants with decreased blood flow to the lungs. These babies, though rare, do not need any medical or surgical therapy during early infancy and should be followed in an out-patient setting until Stage II.

#### 10.1.4. Inter-Atrial Obstruction

Inter-atrial obstruction is unlikely to be an issue in DILV babies with two normal atrioventricular valves and in babies with one common atrioventricular valve emptying into the single ventricle. However, if one of the atrioventricular valves is atretic or markedly hypoplastic/stenotic, adequate-sized atrial septal communication is necessary. If clinical features indicative of obstruction to systemic venous or pulmonary venous return manifest, or if echocardiographic signs of a small-sized patent foramen ovale by two-dimensional study or by increased Doppler flow velocity flow across it suggestive of obstruction of the patent foramen ovale are seen, balloon atrial septostomy [29,30] should be performed.

In babies with inter-atrial obstruction and increased pulmonary blood flow, though such a combination is not uncommon [31], a quick and expected reduction in pulmonary vascular resistance occurs following relief of restrictive patent foramen ovale, whether such a procedure is performed by transcatheter methods or by surgical septectomy [31]. Hence, we recommended that banding of the pulmonary artery [28] be performed with no hesitancy at the time of relief of obstruction across the atrial septum so that there is a good probability of increasing the chances of controlling congestive heart failure, decreasing the pulmonary artery pressures, and avoiding development of pulmonary vascular obstructive changes.

#### 10.1.5. Inter-Ventricular Obstruction

Inter-ventricular obstruction secondary to constriction of BVF may be present when the infants initially present as neonates or may develop subsequently [5], as mentioned above. Such a diminution in the size of the BVF results in sub-pulmonary obstruction in babies without TGA (rare) resulting in decreased blood flow to the pulmonary circuit or subaortic obstruction in babies with TGA. In cases with sub-pulmonary obstruction, the treatment is the same as that described in the section on “Reduced Pulmonary Blood Flow”. In subjects with subaortic obstruction, the narrowing is usually bypassed by a procedure described by Damus, Kaye, and Stansel [32,33,34], commonly described as the DKS procedure. This procedure consists of connecting the separated pulmonary artery stump either directly to the aorta or via a Gore-Tex graft [32,33,34]. The pulmonary circuit is perfused by performing a Blalock–Taussig shunt at the same time. Most frequently, the DKS is accomplished in place of direct enlargement of the BVF. The DKS is performed at initial presentation (Stage I) or while performing Stage II or Stage III, as and when BVF constriction is recognized.

#### 10.1.6. Aortic Arch Obstruction

Obstruction of the aortic arch may happen in the form of an interrupted aortic arch or coarctation of the aorta.

##### Aortic Arch Interruption

Babies with aortic arch interruption have a complete lack of continuity between the aortic arch and the descending aorta. This lesion is categorized into three types, namely, A, B, and C on the basis of where the interruption occurs [35,36]. The treatment of an interrupted aortic arch is by rapid intravenous infusion of PGE_1_ to maintain ductal patency and restore perfusion to systemic circuit. After stabilization of the patient, an end-to-end anastomosis of the interrupted aortic segments along with removal of the ductal tissue should be performed [35,37].

##### Aortic Coarctation

In aortic coarctation, there is constriction of the descending aorta distal to the left subclavian artery around the site of the patent arterial duct [38]. Initial management is with intravenous infusion of PGE_1_ to open the ductus arteriosus, effectively bypassing the coarcted segment. Long segment coarctations (Figure 5) require surgical therapy, most commonly by resection of coarcted segment and re-anastomosis of resected aortic segments, end-to-end [39,40], or by extended end-to-end arch aortoplasty [41] as deemed appropriate [39,40,41]. Discrete coarctations may be treated by transcatheter balloon dilatation [42,43,44] or by surgery [39,40,41], largely based on institutional preferences.

### 10.2. Stage II

Regardless of the kind of palliative procedure (s) performed early on, end-to-side joining the superior vena cava with the right pulmonary artery surgically, i.e., the bidirectional Glenn procedure [45] (Figure 11 and Figure 12), is performed roughly at an age of 6 months. If a previous Blalock–Taussig shunt exists, it is disconnected at the same time. Although undertaking this Stage II procedure at 6 months of age is commonly agreed upon, the bidirectional Glenn procedure may be undertaken close to 3 months, provided that normal pulmonary artery pressure and anatomy can be demonstrated. If other abnormalities (see the Inter-stage Issues section below) are detected, they are also addressed while performing the bidirectional Glenn procedure.

In children with persistent left superior vena cava, a bilateral bidirectional Glenn procedure (Figure 13) is accomplished mainly in subjects who have no left innominate vein or if it is narrow. A similar bidirectional Glenn shunt can also be undertaken in children with infrahepatic interruption of the inferior vena cava, irrespective of its association with azygos or hemiazygos continuation, and such surgery is known as the Kawashima operation.

Before performing the bidirectional Glenn operation, one should ensure that the pulmonary artery pressures are within normal limits and that the branch pulmonary arteries are of acceptable size. This is to ensure adequate forward flow across the Glenn shunt since there is no pumping chamber. Such assessment was performed in the past with the use of cardiac catheterization and cine-angiographic studies. Recently, some institutions have been employing echocardiographic, magnetic resonance imaging, or computed tomography studies to achieve such assessment. If pulmonary arteries are stenosed, they may be treated with balloon dilatation or stent placement, as thought to be suitable. Alternatively, repair of pulmonary arteries may be performed while undertaking the bidirectional Glenn procedure. 

### 10.3. Inter-Stage Issues

In children with one ventricle, the systemic and pulmonary circuits work in parallel in place of the normal in-series circulation and a subtle balance among the two circuits should be preserved in order to maintain satisfactory perfusion of both systemic and pulmonary circulations [15,46]. If such a balance cannot be maintained, substantial morbidity and even mortality may occur in these susceptible infants. The reported mortality in between the stages varied between 5 and 15% [15,47,48]. Some studies have detected the causes of inter-stage mortality [48,49]. The identified problems are obstructed patent foramen ovale, aortic arch obstruction, pulmonary artery stenosis or distortion, atrioventricular valve regurgitation, blockage of the aorta-pulmonary shunts, and inter-current illnesses [47,49]. The mortality in between the stages occurs more frequently between Stage I and Stage II than between Stage II and Stage III. While these studies [47,49] largely involve hypoplastic left heart syndrome patients, such data are similarly pertinent to other single ventricle patients including DILV [14,15,48]. Policies to prevent and treat the inter-stage issues are intermittent clinical evaluation along with echo-Doppler studies (and when necessary magnetic resonance imaging and computed tomography studies) to identify the abnormalities listed above and offer satisfactory treatment of the identified issues in an attempt to avert/lessen the morbidity and mortality [14,15,48,50]. Careful periodic clinical and echocardiographic evaluation is most likely to accomplish this goal. Inter-current illnesses which result in dehydration, disturbances in acid–base status, or elevated temperature may disrupt the balance among the pulmonary and systemic circuits and the babies may develop critical illnesses [47,48]. Prompt attention for addressing inter-current illnesses is germane [15,48]. A detailed discussion of how to address the inter-stage problems is beyond the scope of this review and the interested persons are requested to review another paper published recently [15].

### 10.4. Stage III

During the final Stage III, the blood from the inferior vena cava is rerouted into the pulmonary artery and a fenestration is created in between the conduit connecting the inferior vena cava with the pulmonary artery and the remaining atrial mass. The author arbitrarily separates the procedure into Stage IIIA, consisting of diversion of the inferior vena caval flow into the pulmonary artery and Stage IIIB during which the fenestration is occluded [12,14].

#### 10.4.1. Stage IIIA

During the Stage IIIA, the TCPC is performed by conveying the inferior vena caval flow into the pulmonary artery either through a lateral tunnel method [51,52] or by an extra-cardiac, non-valved Gore-Tex conduit [53,54] (Figure 14, Figure 15, Figure 16 and Figure 17). This surgery is generally undertaken between one and two years of age, commonly one year following the bidirectional Glenn procedure. At this time, the majority of surgeons appear to be favoring an extra-cardiac non-valved conduit to complete the last stage of Fontan surgery. Most surgeons seem to prefer creation of a fenestration, 4–6 mm in diameter, in between the conduit and the remaining atrial mass [55,56] (Figure 16 and Figure 17). Whereas creating a fenestration while performing the Fontan surgery was initially recommended for patients with increased risk for Fontan surgery failure [55,56], most pediatric cardiac surgeons and pediatric intensivists favor fenestration because creating a fenestration while performing the Fontan surgery reduces the mortality rates and decreases the postoperative morbidity [12,14].

#### 10.4.2. Stage IIIB

Stage IIIB is usually undertaken 6–12 months after Stage IIIA of the Fontan surgery; the fenestration is generally occluded (Figure 18 and Figure 19) by transcatheter methods [12,55,56,57,58,59]. Most devices utilized in the past to close atrial septal defects [55,57,58,59] were employed for fenestration closure. Nevertheless, at the present time, Amplatzer Septal Occluders (St. Jude Medical, Inc., St. Paul, MN—Abbott) are the most commonly used devices to achieve fenestration occlusion. Immediately prior to closure of the fenestration, it should be test occluded [57] to ensure that the patient tolerates the procedure. If there are other shunts across the conduit, they are also addressed by transcatheter occlusion.

## 11. Post-Fontan Follow-Up

Subsequent to Fontan completion, clinical follow-up in an outpatient setting is essential; it is recommended that the patients are assessed at 1-, 6-, and 12-months following Stage IIIB Fontan and yearly afterwards [12,13,14,15]. Inotropic and/or diuretic medications may be given as necessary. Captopril or Enalapril (angiotensin-converting enzyme inhibitors) may be administered to institute afterload reduction. Anticoagulation is provided with platelet-inhibiting drugs such as aspirin (2–5 mg/kg/day) in pediatric-age patients and Clopidogrel (75 mg/day) in adults. This drug therapy is instituted to prevent thrombosis, instead of Warfarin. Such an approach is largely based on the data of a multicenter, randomized clinical trial [60] which showed no substantial variance between Aspirin and Warfarin regimens.

Several complications were noticed during the follow-up of the Fontan patients and these include development of arrhythmias, onset of obstruction within the Fontan pathways, residual defects producing shunts, thrombus formation, embolism causing episodes of cerebro-vascular accidents or transient ischemic attacks, cyanosis due to right to left shunting, formation of collateral vessel connections between systemic and pulmonary venous circuits, systemic venous congestion, and protein-losing enteropathy [12,13,61]. Fortunately, these complications seem to occur less frequently in patients who are treated with the currently used staged TCPC with extra-cardiac Gore-Tex conduit than patients who had older types (atrio-pulmonary connection) of Fontan operations. During the follow-up evaluation, the caregiver should watch for development of these complications and address them as soon as they are detected. Detailed clinical, echocardiographic, and cardio-pulmonary exercise testing [62] evaluations at the above-mentioned intervals should be performed and, if necessary, other imaging studies should be secured. Discussion of the management of these complications is beyond the scope of this presentation and the interested physicians are referred to other papers on this topic published elsewhere [12,14,61,63]. Most patients do well, but an occasional patient may require other forms of therapy: namely, decreasing the pressure in the conduit by making a fenestration in between the Gore-Tex conduit and the atrial mass, changing the atrio-pulmonary type of Fontan to TCPC, initiating sequential atrio-ventricular pacing [13,14,15], and cardiac transplantation [64,65,66].

## 12. Summary and Conclusions

Univentricular hearts are rare complex CHDs and constitute less than 2% of all CHDs. DILV is the most common among univentricular atrioventricular connections and is discussed in this review. In this anomaly, there is one functioning ventricle which has left ventricular morphology (most of the cases), receives both atrioventricular valves, and gives origin to an outlet chamber with morphologic features of a right ventricle. The great vessels are frequently transposed and pulmonary stenosis is present in the majority of patients. Some of these patients have other obstructive lesions including BVF constriction and coarctation of the aorta. DILV and associated defects can be defined by echo-Doppler studies with occasional need for other imaging studies. Therapy is largely based on the type of associated cardiac defects and the degree of hemodynamic abnormality that they produce and were detailed in the review. These children are commonly treated with staged TCPC (Fontan) in three stages. Stages I provides palliation of the hemodynamic abnormalities detected at presentation, Stage II consists of the bidirectional Glenn procedure, and Stage III is TCPC with an extracardiac conduit with fenestration; the latter is subsequently closed by transcatheter methodology. The prevalence of inter-stage mortality is high (5% to 15%); it is more frequent between Stages I and II than between Stages II and III. The inter-stage mortality is largely related to: restrictive atrial communication; obstruction of the aortic arch; distortion/stenosis of the pulmonary arteries; atrioventricular valve insufficiency; shunt blockage; and inter-current illnesses. Intermittent clinical evaluation along with echo-Doppler or other imaging studies to identify these anomalies should be methodically pursued and when detected, they should be promptly addressed. Even minor inter-current illnesses must be treated aggressively. Periodic follow-up after the Fontan procedure while administering medications in some patients and vigilance to detect complications of the Fontan surgery is necessary. It was concluded that DILV can be successfully diagnosed with the currently available non-invasive techniques and the defect can effectively be managed with the presently available medical, catheter interventional, and surgical therapeutic methodologies.

## Figures and Tables

**Figure 1 children-09-01274-f001:**
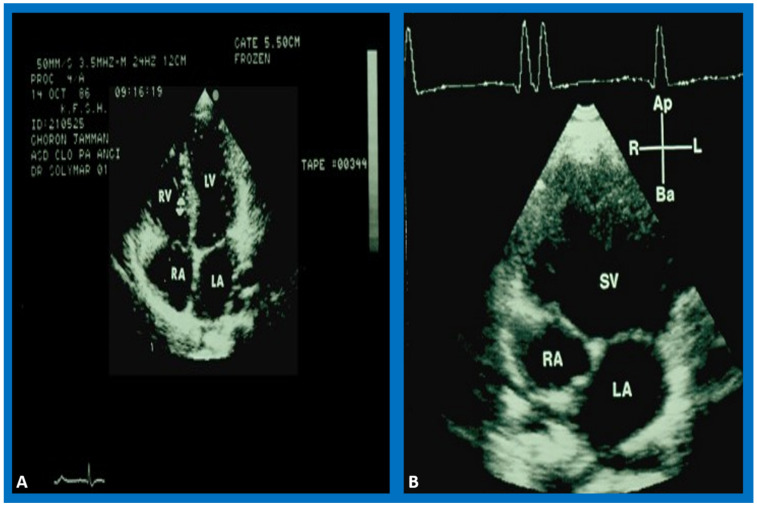
Selected video frames from two different patients. (**A**) shows the ventricular septum between the ventricles in a normal child while (**B**) demonstrates the absence of the ventricular septum. The figures show top-to-bottom reversal; this is because these figures were prepared prior to the American Society of Echocardiography recommendation of the current way in which we display the images. Left atrium (LA), left ventricle (LV), right atrium (RA), right ventricle (RV), and single ventricle (SV) are labeled.

**Figure 2 children-09-01274-f002:**
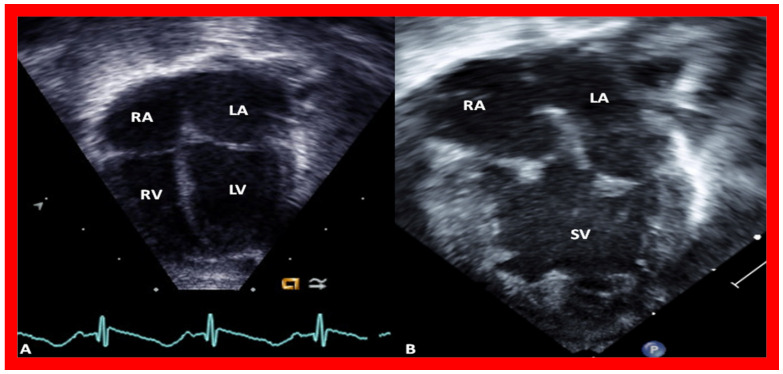
The 2D echo images from apical four-chamber projections of two children, one with two ventricles (**A**) and another with a single ventricle (SV) (**B**). These figures were prepared after the American Society of Echocardiography recommendations of the current way in which we display the images. Left atrium (LA), left ventricle (LV), right atrium (RA), and right ventricle (RV) are marked. Reproduced from Reference [4].

**Figure 3 children-09-01274-f003:**
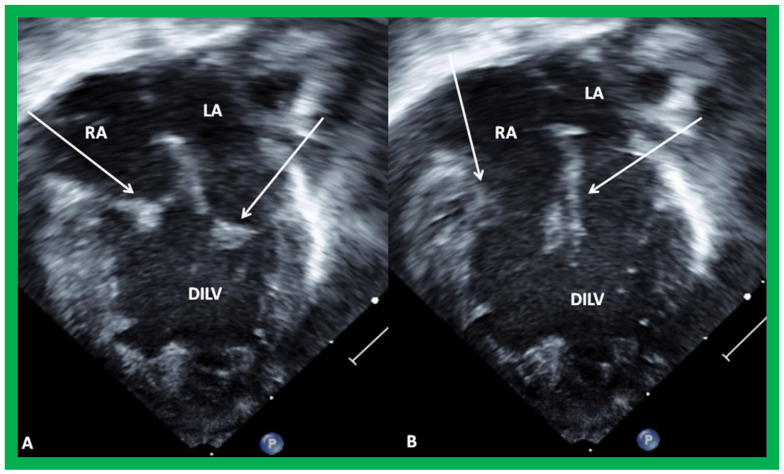
The 2D echo images from apical four-chamber projections of a child who was diagnosed to have double inlet left ventricle (DILV). (**A**) shows closed and (**B**) shows open atrioventricular valves as indicated by arrows. No evidence for a ventricular septum is seen. The right ventricle is not imaged in this projection (see Figure 4). Left atrium (LA) and right atrium (RA) are labeled. Reproduced from Reference [3].

**Figure 4 children-09-01274-f004:**
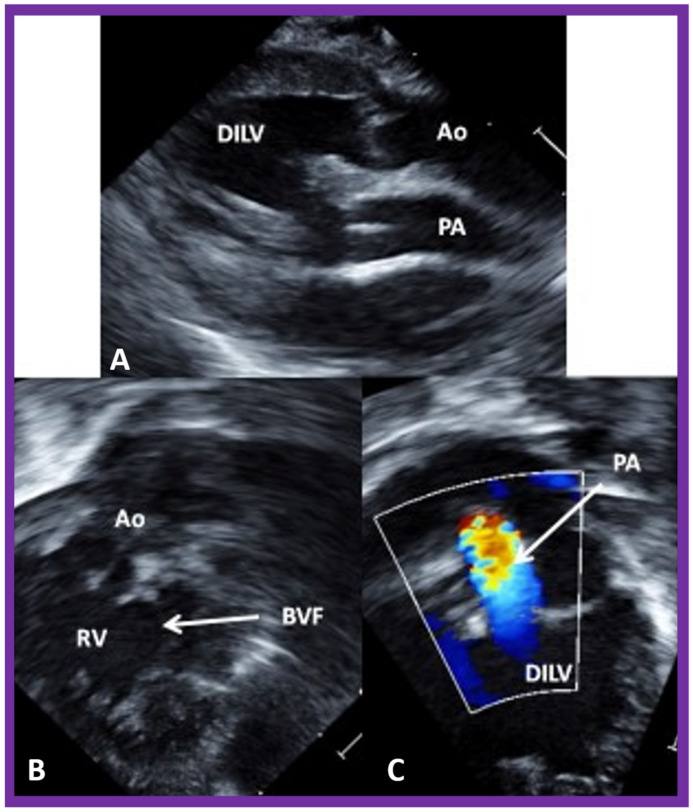
The 2D echo/color Doppler images from the parasternal long axis (**A**) and modified apical (**B**,**C**) projections of the case illustrated in Figure 3, indicating transposed great arteries (**A**). Note the anterior location of the aorta (Ao) and posterior location of the pulmonary artery (PA) (**A**). The connection of the right ventricular (RV) chamber with the double-inlet left ventricle (DILV) via a bulbo-ventricular foramen (BVF) is seen in (**B**). This RV provides an origin to the aorta (Ao) as shown in (**B**). The pulmonary artery (PA) arises from the main ventricle, DILV, as seen in (**C**). Turbulent flow in the PA is seen (**C**) and is suggestive of obstruction. Continuous wave Doppler (not illustrated) demonstrated increased Doppler velocity indicative of severe pulmonary narrowing. Reproduced from Reference [3].

**Figure 5 children-09-01274-f005:**
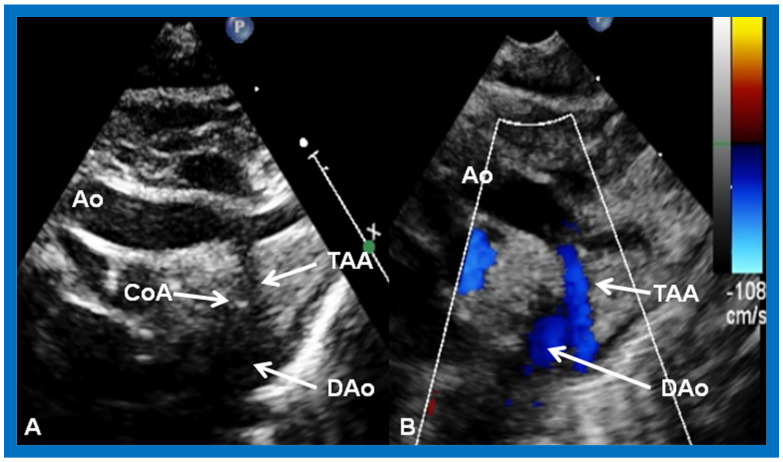
Selected video images secured from the suprasternal notch view of the arch of the aorta (Ao) in two-dimensional (**A**) and color Doppler (**B**) pictures demonstrating coarctation of the aorta (CoA) and hypoplastic transverse aortic arch (TAA). Descending aorta (DAo) is labeled. Reproduced from Reference [6].

**Figure 6 children-09-01274-f006:**
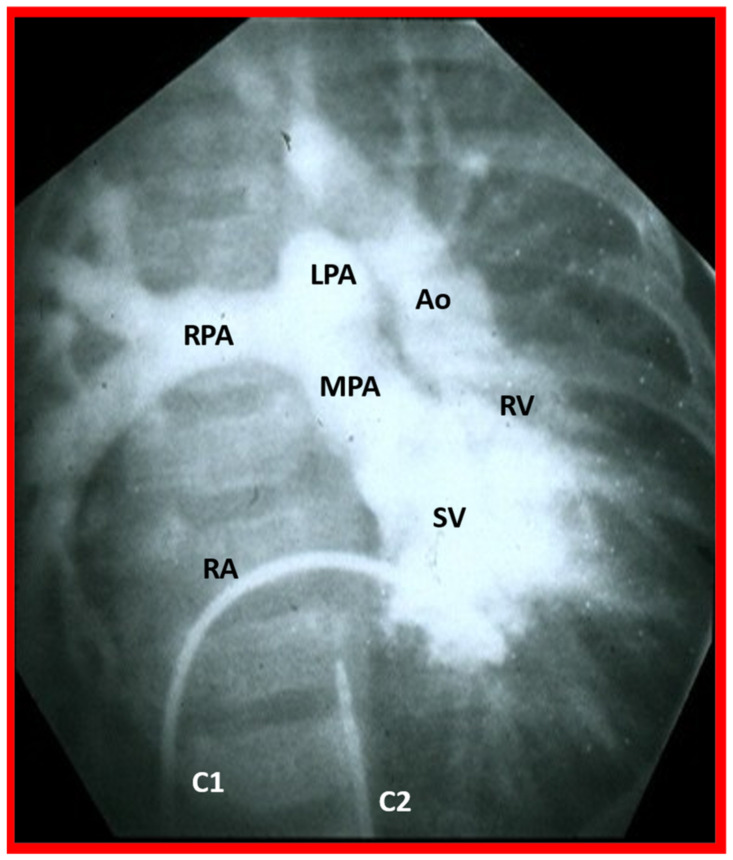
Selected cine frame in the postero-anterior projection of a single ventricular (SV) cine-angiogram demonstrating simultaneous opacification of the main (MPA), left (LPA), and right (RPA) pulmonary arteries from the SV and the aorta (Ao) from the right ventricle (RV). Note that the Ao is positioned to the left of the MPA, indicating l-transposition of the great vessels. C1. Catheter in the inferior vena cava (not marked) which was advanced into the right atrium (RA) and then into the SV; C2. Catheter in the descending aorta (not marked).

**Figure 7 children-09-01274-f007:**
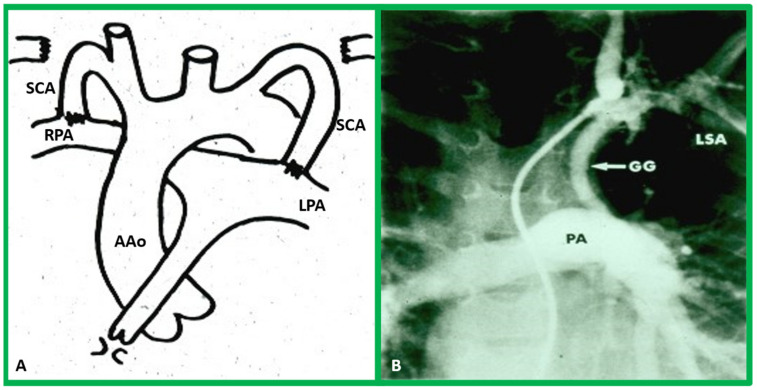
(**A**) Hand drawing (drawn by Dr. Taussig herself) illustrating the concept of the Blalock–Taussig shunt [25] showing the anastomosis of the subclavian arteries (SCAs) to the left (LPA) or right (RPA) pulmonary arteries, respectively. (**B**) Selected frame from a cine-angiogram of the Gore-Tex graft (GG) showing modified Blalock–Taussig (BT) shunts [18]. This image demonstrates widely patent BT shunt and excellent visualization of the pulmonary artery (PA). AAo, ascending aorta; LSA, left subclavian artery.

**Figure 8 children-09-01274-f008:**
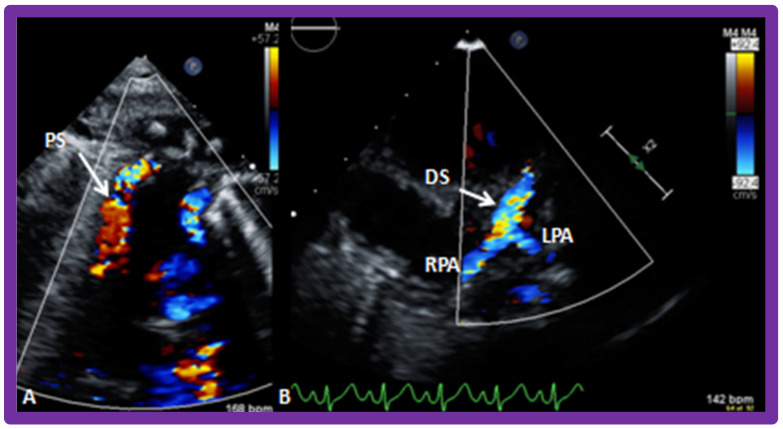
Selected video images secured with the transducer positioned in the suprasternal notch illustrating the proximal part of the shunt (PS) with color Doppler (**A**). In another transducer angulation (**B**), the Doppler flow from the distal portion of the shunt (DS) into both the right (RPA) and left (LPA) pulmonary arteries is imaged. Reproduced from Reference [26].

**Figure 9 children-09-01274-f009:**
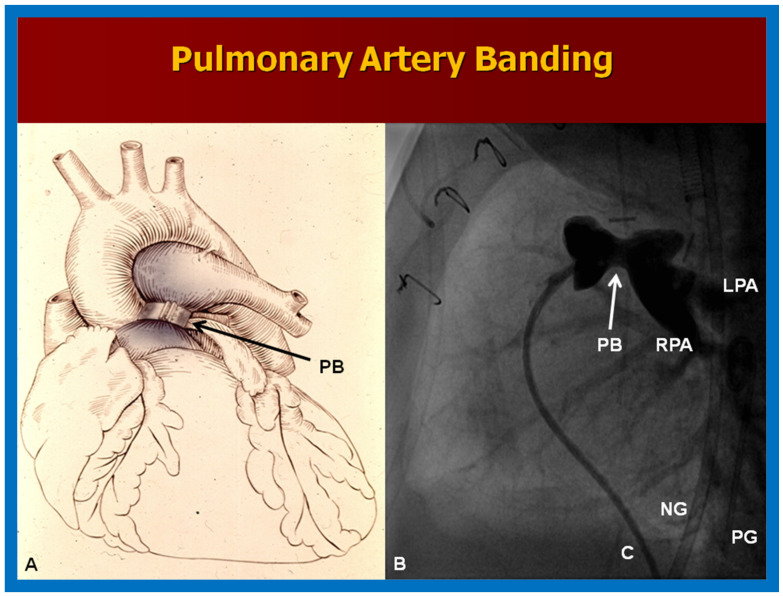
(**A**) Pictorial depiction of pulmonary artery banding (PB) for babies with severely augmented blood flow into the lungs and heart failure. (**B**) Cine image from a pulmonary artery cine-angiogram in conventional lateral projection illustrating the narrowed segment of the pulmonary artery (PB) indicated by an arrow in a baby who underwent PB. The catheter (C), left pulmonary artery (LPA), nasogastric tube (NG), pigtail catheter (PG), and right pulmonary artery (RPA) are labeled.

**Figure 10 children-09-01274-f010:**
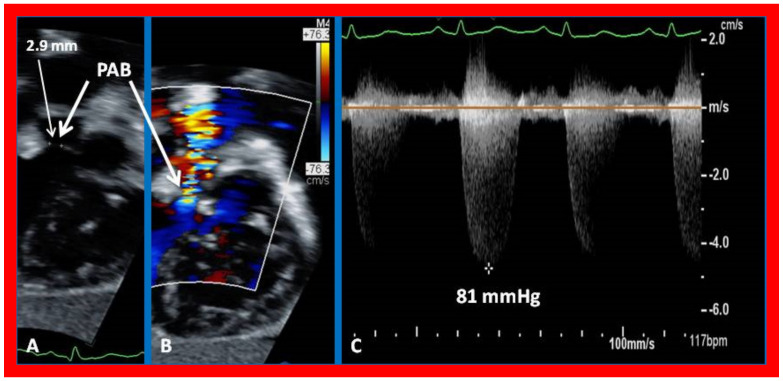
Selected echo-Doppler images illustrating the pulmonary artery band (PAB). Note, narrow PAB diameter by two-dimensional echo in (**A**) and by color Doppler imaging in (**B**). Continuous wave Doppler reveals a significant gradient (81 mmHg) across the PAB (**C**). Reproduced from Reference [26].

**Figure 11 children-09-01274-f011:**
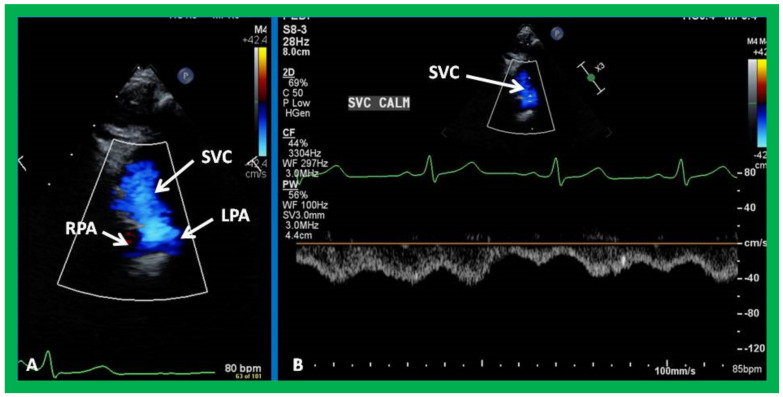
Echo images secured with the transducer positioned in the suprasternal notch illustrating the bidirectional Glenn shunt. Note that the superior vena cava (SVC) is draining into the right (RPA) and left (LPA) pulmonary arteries shown by color Doppler (**A**). Low Doppler flow velocities through the bidirectional Glenn shunt (**B**) indicate that the shunt is not obstructed. Reproduced from Reference [26].

**Figure 12 children-09-01274-f012:**
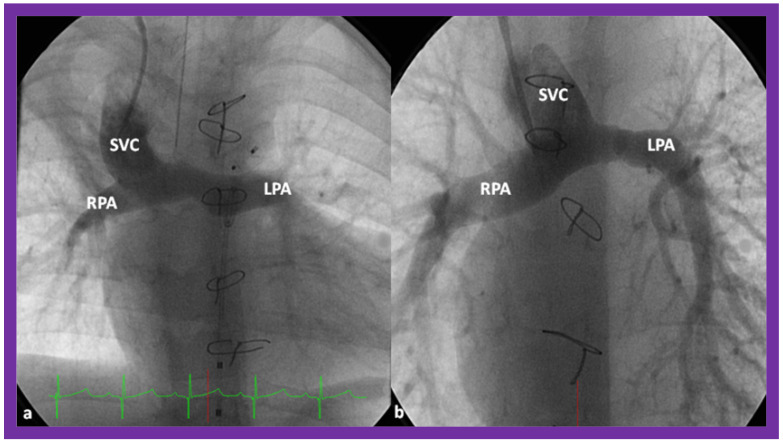
Cine-angiographic images illustrating bidirectional Glenn procedure in two separate children (**a**,**b**). The unimpeded flow of blood from the superior vena cava (SVC) to the right (RPA) and left (LPA) pulmonary arteries is demonstrated. Sternal wires related to previous surgical procedures are shown and are not labeled. Reproduced from Reference [12].

**Figure 13 children-09-01274-f013:**
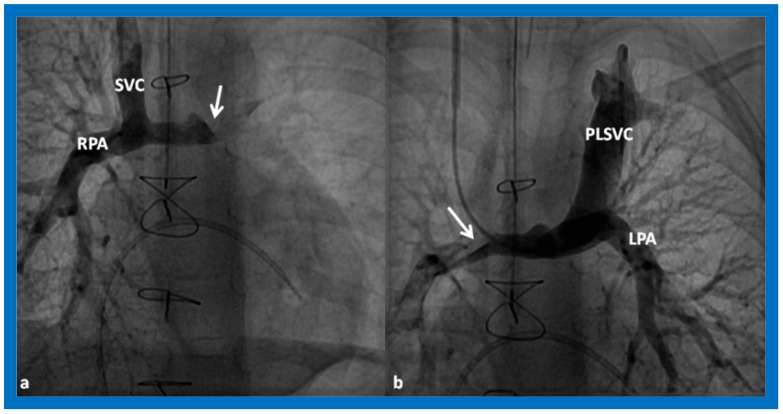
Selected cine-angiographic images demonstrating the bilateral bidirectional Glenn procedure. (**a**) This is an angiogram with contrast injection into the right superior vena cava (SVC) illustrating rapid visualization of the right pulmonary artery (RPA). The unopacified blood from a persistent left superior vena cava (PLSVC) is shown with an arrow in “(**a**)”. (**b**). This is an angiogram from the PLSVC demonstrating rapid visualization of the left pulmonary artery (LPA). The unopacified blood from the right SVC is shown with an arrow in “(**b**)”. These figures demonstrate flow of the contrast material from both the SVCs into the LPA and RPA, respectively, without obstruction. Reproduced from Reference [12].

**Figure 14 children-09-01274-f014:**
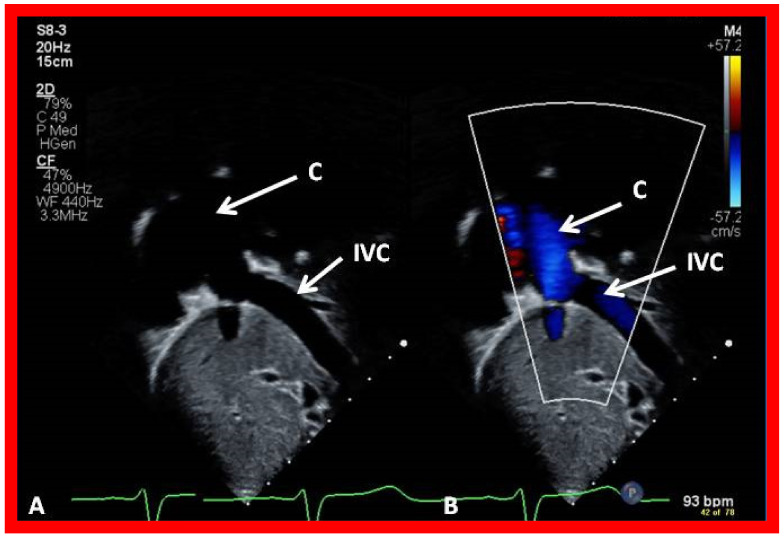
Echo images illustrating anastomosis of the inferior vena cava (IVC) with the conduit (C) by 2D (**A**) and color flow (**B**) imaging. Note widely patent IVC–C junction both by 2D (**A**) and by laminar flow with color Doppler (**B**). Reproduced from Reference [26].

**Figure 15 children-09-01274-f015:**
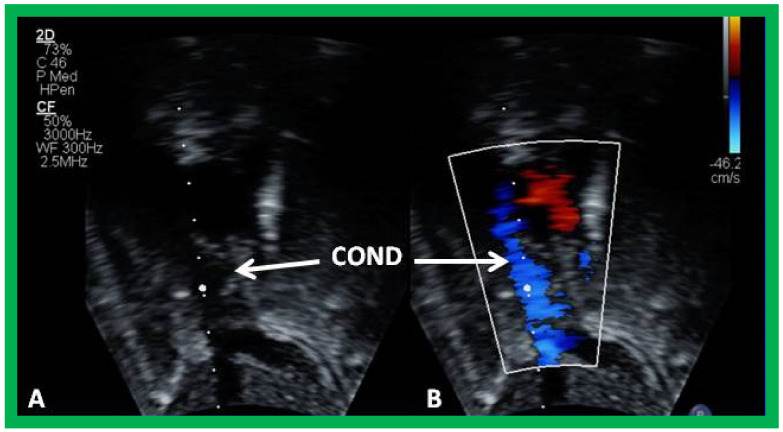
Echo images illustrating widely open conduit (COND) by 2D (**A**) and color flow imaging (**B**). Note laminar flow in B which suggests a lack of obstruction in the COND. Reproduced from Reference [26].

**Figure 16 children-09-01274-f016:**
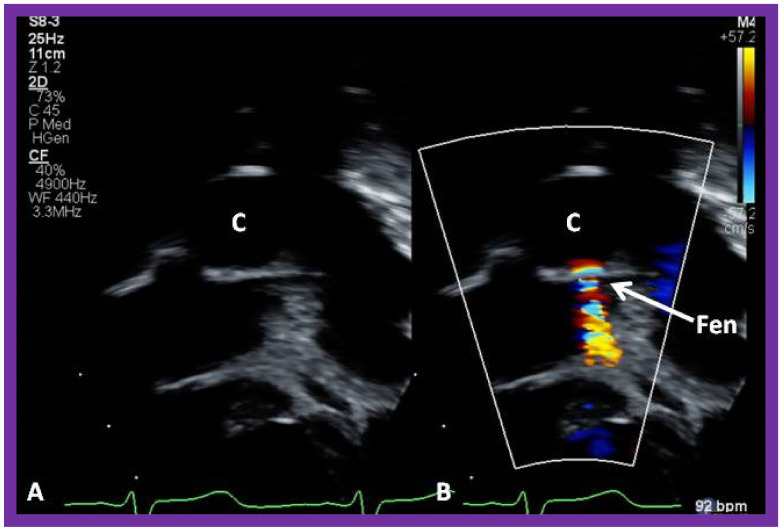
Echo images from an apical four chamber projection by 2D (**A**) and color flow (**B**) imaging demonstrating cross-sectional images of the conduit (C) in (**A**,**B**) and a fenestration (Fen) in (**B**). Note that the flow across the Fen is turbulent (**B**). Reproduced from Reference [26].

**Figure 17 children-09-01274-f017:**
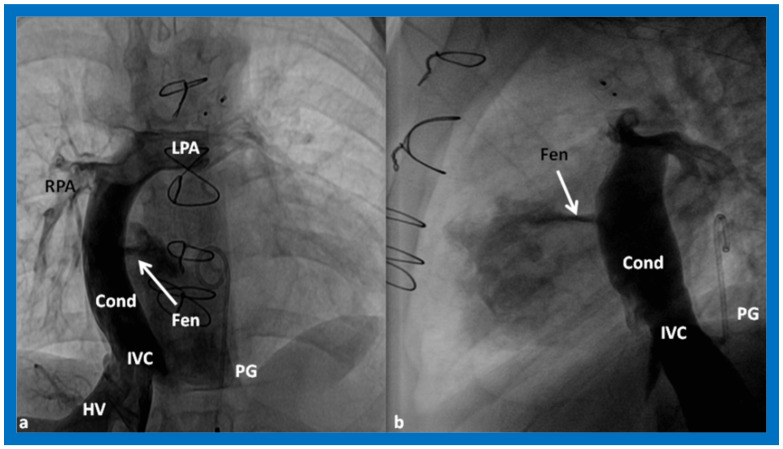
Cine-angiographic images in postero-anterior (**a**) and lateral (**b**) views, illustrating Stage IIIA of the Fontan operation rerouting the blood flow from the inferior vena cava (IVC) into the right (RPA) and left (LPA) pulmonary arteries through a non-valved Gore-Tex conduit (Cond). The flow of the contrast material via the fenestration (Fen) is indicated by the arrows in both **a** and **b**. HV, hepatic veins; PG, pigtail catheter. Modified from Reference [12].

**Figure 18 children-09-01274-f018:**
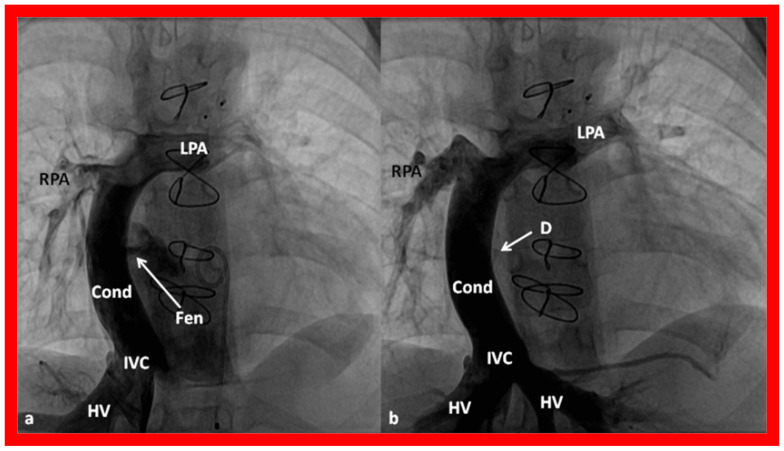
Cine-angiographic images in antero-posterior projection, illustrating Stage IIIA Fontan operation transmitting blood flow from the inferior vena cava (IVC) into the pulmonary arteries through a non-valve Gore-Tex conduit (Cond). The fenestration (Fen) is indicated by an arrow in (**a**). The Fen was occluded with an Amplatzer device (D), again marked by an arrow in (**b**) (Stage IIIB). The hepatic veins (HV), left pulmonary artery (LPA), and right pulmonary artery (RPA) are labeled. Reproduced from Reference [12].

**Figure 19 children-09-01274-f019:**
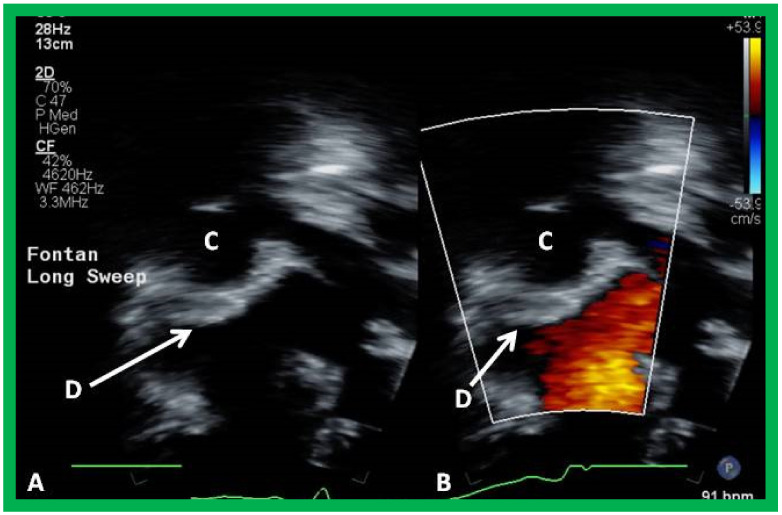
Echo images from apical four chamber projection illustrating the location of the Amplatzer device (D), indicated by arrows in (**A**,**B**). Note that there is no residual shunt demonstrated as shown in B. Conduit (C) is labeled. Reproduced from Reference [26].

## Data Availability

Not applicable.

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
