# Peer review of "Double-Inlet Left Ventricle"

_children, 2022, doi:10.3390/children9091274_

Round 1
Reviewer 1 Report
I would like to congratulate the author with submitted manuscript: "Double-Inlet Left Ventricle. Review".
I find the manuscript very interesting and presented in comprehensive way including pathophysiology, clinical features, results of diagnostic tool and presented therapeutic options.
I strongly vote for acceptance.
Kind regards
Author Response
The reviewer states that "I would like to congratulate the author with submitted manuscript: "Double-Inlet Left Ventricle. Review". I find the manuscript very interesting and presented in comprehensive way including pathophysiology, clinical features, results of diagnostic tool and presented therapeutic options. I strongly vote for acceptance." The author thanks for the reviewer for the comments. No revisions are recommended. Please convey my thanks to the reviewer.
Reviewer 2 Report
Author’s contributions:
The author has made a review of methods for detection and diagnosis of Double-inlet left ventricle, which is a birth defect in the heart. After a brief description of the problem, an extended review has been made of appropriate methods for detection of this defect in heart.
According to the review, the great vessels are often transposed and pulmonary stenosis is seen two-thirds of patients. The author has defined that the anatomy and pathophysiology can be defined by Echo-Doppler studies with rare need for other methods for image analysis.
The main contribution of the author is the summary that Double-inlet left ventricle can be effectively diagnosed with Echo-Doppler studies and this heart defect can be successfully treated with the currently existing medical, catheter interventional and surgical treatment practices.
I have some reviewer notes:
There are many abbreviations. It will be good to add list of abbreviation.
Introduction part. At the end of this part it will be good to summarize how the paper is structured. Also, the aim of the work has to be clearly stated.
Pathologic Anatomy part. “Obstruction at the sub-aortic region has been found in subjects with TGA and is related to the small size of the ventricular septal defect; more correct terminology is bulbo-ventricular foramen.”, needs a brief explanation, why the available terminology is not correct?
Pathophysiology part. “In infants without Pulmonary stenosis, the pulmonary blood flow is not increased at and shortly after birth because of high neonatal pulmonary vascular resistance.” IS there a connection with parent or relatives of the child? Needs brief description.
Clinical Features part. “Infants with significant PS present with symptoms of cyanosis, tachypnea and tachycardia.” Needs a brief description of the reasons.
Chest X-ray, Electrocardiogram (ECG), Echocardiogram, magnetic resonance imaging (MRI), computed tomography (CT), selective cineangiography (SC). After these sections you have to make a comparative analysis of the methods. What are their advantages and limitations?
Page 12. “Before performing the bidirectional Glenn procedure, one must ensure that the PA pressures are within normal limits and that the branch PAs are of acceptable size.” You have to give a short description why we have to do that.
Page 12. “Policies to prevent and treat the inter-stage issues are intermit-tent clinical evaluation along with Echo-Doppler and other imaging studies to detect the abnormalities described above and offer satisfactory relief of detected issues in an attempt to prevent/reduce the morbidity and mortality.” Which of these methods have potential for early diagnosis of the problem?
Page 16. “During follow-up evaluation, the care-giver should watch for development of these complications and address them as soon as they are detected.” Needs more explanation.
Summary and Conclusion part. It will be good to comment in which cases, the methods for diagnosis are appropriate. Also in which cases other methods can overcome their limitations.
The references are not cited in the text. It is not clear, where are all commented findings from.
The whole paper has to be formatted according to the journal requirements.
I have some suggestions:
Format the paper according to the journal requirements. Make more comparative analyses.
Author Response
The reviewer nicely summarizes the paper, thanks. No response is necessary.
The reviewer states “There are many abbreviations. It will be good to add list of abbreviation.” – Since the journal format does not have a section for including a list of observations, I have reduced the number of abbreviations during the process of revision.
The reviewer suggests “Introduction part. At the end of this part, it will be good to summarize how the paper is structured. Also, the aim of the work has to be clearly stated.” – Since there is an ‘Abstract’ in the beginning of the paper as well as ‘Summary and Conclusion’ at the end of the paper, I believe it is superfluous to provide a summary in the ‘introduction section’. I have however, added aims of the paper, as suggested by the reviewer.
The reviewer asks “Pathologic Anatomy part. “Obstruction at the sub-aortic region has been found in subjects with TGA and is related to the small size of the ventricular septal defect; more correct terminology is bulbo-ventricular foramen.”, needs a brief explanation, why the available terminology is not correct?” - An explanation re terminology is now added.
The reviewer questions “Pathophysiology part. “In infants without Pulmonary stenosis, the pulmonary blood flow is not increased at and shortly after birth because of high neonatal pulmonary vascular resistance.” Is there a connection with parent or relatives of the child? Needs brief description.” - All babies have high pulmonary vascular resistance at birth. I do not believe modification of the text here is necessary.
The reviewer suggests “Clinical Features part. “Infants with significant PS present with symptoms of cyanosis, tachypnea and tachycardia.” Needs a brief description of the reasons.” – Done.
The reviewer recommends “Chest X-ray, Electrocardiogram (ECG), Echocardiogram, magnetic resonance imaging (MRI), computed tomography (CT), selective cineangiography (SC). After these sections you have to make a comparative analysis of the methods. What are their advantages and limitations?” – A short section is added as recommended by the reviewer.
The reviewer suggests “Page 12. “Before performing the bidirectional Glenn procedure, one must ensure that the PA pressures are within normal limits and that the branch PAs are of acceptable size.” You have to give a short description why we have to do that.” – Done, as suggested.
The reviewer comments “Page 12. “Policies to prevent and treat the inter-stage issues are intermit-tent clinical evaluation along with Echo-Doppler and other imaging studies to detect the abnormalities described above and offer satisfactory relief of detected issues in an attempt to prevent/reduce the morbidity and mortality.” Which of these methods have potential for early diagnosis of the problem?” – clinical evaluation and echo-Doppler studies – added as recommended by the reviewer.
The reviewer recommends “Page 16. “During follow-up evaluation, the care-giver should watch for development of these complications and address them as soon as they are detected.” Needs more explanation.” – referenced.
The reviewer suggests “Summary and Conclusion part. It will be good to comment in which cases, the methods for diagnosis are appropriate. Also, in which cases other methods can overcome their limitations.” – Revised accordingly.
The reviewer suggests “The references are not cited in the text. It is not clear, where are all commented findings from.” – During the process of the revision, I have carefully gone over the reference citations and made changes when deemed appropriate.
The reviewer comments “The whole paper has to be formatted according to the journal requirements.” – I beg to disagree; the paper is formatted as per the journal’s requirements.
Reviewer 3 Report
1 According to the manuscript from Dr. Khairy (PMID: 17296869), HLHS is the most common single ventricle disease, not DILV
Author Response
The reviewer comments: "According to the manuscript from Dr. Khairy (PMID: 17296869), HLHS is the most common single ventricle disease, not DILV" - I agree. During the process of the revision, I will ensure that the manuscript does not suggest that the most common single ventricle disease is not DILV.
Please convey my thanks to the reviewer.
Round 2
Reviewer 2 Report
The paper is corrected according to the reviewer notes.